# Teacher–Student Behavior Recognition in Classroom Teaching Based on Improved YOLO-v4 and Internet of Things Technology

**Henghuai Chen [1] and Jiansheng Guan [2,*]**

1   Organization Department of the Party Committee, Xiamen University of Technology, Xiamen 361024, China
2   School of Electrical Engineering and Automation, Xiamen University of Technology, Xiamen 361024, China
*   Correspondence: jsguan@xmut.edu.cn

**Abstract:** Based on the classroom teaching scenarios, an improved YOLO-v4 behavior detection algorithm is proposed to recognize the behaviors of teachers and students. With the development of CNN (Convolutional Neural Networks) and IoT (Internet of Things) technologies, target detection algorithms based on deep learning have become mainstream, and typical algorithms such as SSD (Single Shot Detection) and YOLO series have emerged. Based on the videos or images collected in the perception layer of the IoT paradigm, deep learning models are used in the processing layer to implement various intelligent applications. However, none of these deep learning-based algorithms are perfect, and there is room for improvement in terms of detection accuracy, computing speed, and multi-target detection capabilities. In this paper, by introducing the concept of cross-stage local network, embedded connection (EC) components are constructed and embedded at the end of the YOLO-v4 network to obtain an improved YOLO-v4 network. Aiming at the problem that it is difficult to quickly and effectively identify the students' actions when they are occluded, the Repulsion loss function is connected in series on the basis of the original YOLO-v4 loss function. The newly added loss function consists of two parts: RepGT loss and RepBox loss. The RepGT loss function is used to calculate the loss values between the target prediction box and the adjacent ground truth boxes to reduce false positive detection results; the RepBox loss function is used to calculate the loss value between the target prediction box and other adjacent target prediction boxes to reduce false negative detection results. The training and testing are carried out on the classroom behavior datasets of teachers and students, respectively. The experimental results show that the average precision of identifying various classroom behaviors of different targets exceeds 90%, which verifies the effectiveness of the proposed method. The model performs well in sustainable classroom behavior recognition in educational context, accurate recognition of classroom behaviors can help teachers and students better understand classroom learning and promote the development of intelligent classroom model.

**Keywords:** deep learning; IoT; behavior recognition; smart classroom; computer vision; YOLO-v4; object detection; embedded connection

## 1. Introduction

At present, with the increasing definition of video recording devices in work and life scenarios, the preconditions for a video-based classroom behavior analysis of teachers and students have been met. The development of artificial intelligence (AI) has reached a new stage. Under the developing trend of emerging technologies and industry demands, multi-domain integration and deep learning (DL) techniques have become new features of AI-based implementations. Classroom behavior analysis aims to study the internal mechanisms of teaching activities of teachers and academic development of students in the sustainable classroom, so as to help teachers and students reflect on their own classroom performance and promote the improvement of sustainable classroom teaching quality [1]. In traditional sustainable classroom teaching, we can only rely solely on

teachers to control and maintain classroom discipline and students' status, which adds a significant burden to teachers in addition to their normal teaching tasks. In addition, most of the traditional classroom teaching behavior analysis methods collect and analyze data through self-evaluation, manual supervision, manual coding and other methods, which have disadvantages such as a strong coding subjectivity and a small sample size and being time-consuming and labor-intensive, resulting in poor interpretability and low scalability [2].

The popularity of AI technology and IoT (Internet of Things) paradigm has brought opportunities to improve these shortcomings. The IoT paradigm can be divided into four layers: perception layer, network layer, processing layer and application layer. The perception layer is used to perceive the physical world and collect various types of information in the real world. The key technologies of this layer are recognition and perception techniques, for example, the high-definition cameras installed in the classroom are used to collect information on teachers and students. The network layer is used for information transmission and includes various types of networks. The key technologies of this layer are short-range wireless communication technology and long-distance technology, such as Zigbee, NFC (Near Field Communication), Bluetooth, Internet, 2G/3G/4G communication network, etc. The processing layer is equivalent to the human brain and plays the role of data storage and processing tasks, including data storage, data management, data mining and data analysis based on various intelligent algorithms. The application layer is directly oriented to users and makes data analysis and early warnings based on sensor information, such as smart classroom applications. Using intelligent technology to collect and analyze data can identify classroom behavior in a more timely and comprehensive manner, gain insight into the status of teachers and students in the sustainable classroom teaching process and provide a powerful tool for improving teaching quality [3].

Compared with traditional methods, target detection algorithms based on Convolutional Neural Networks (CNN) have gradually become a new research focus. CNN-based deep learning models improve the learning quality by increasing the number of convolutional layers [4]. In general, object detection based on deep learning performs better than detection methods based on handcrafted features. From a technical point of view, CNN uses the back-propagation algorithm for feedback learning, and through automated feature extraction and multi-layer convolution learning, it minimizes human intervention and improves modelling capabilities [5]. In addition, data enhancement techniques have been utilized in recently proposed CNN networks to increase the quality and diversity of learning samples, which is beneficial to the improvement of the detection accuracy [6]. Deep-learning-based Target detection algorithms have been applied to various scenarios such as pedestrian re-identification [7], license plate detection [8] and power line inspection [9], etc.

There are two main categories of target detection methods based on deep learning: firstly, the two-stage target detection algorithms based on region proposals, such as R-CNN, Fast R-CNN and Faster R-CNN. Secondly, regression-based one-stage target detection algorithms, such as YOLO, RetinaNet and EfficientDet, among which the YOLO series are the most widely used one-stage detection algorithms [10]. In May 2016, Redmon et al. [11] proposed the first regression-based target detection method YOLO-v1, which received extensive attention from researchers as soon as it came out. With the development of the YOLO family, in 2020, it was updated to the fourth generation. The experimental results on the standard dataset prove that YOLO-v4 is one of the best target detection algorithms at present, and the fast detection speed is the distinguishing feature of YOLO-v4 [12].

The YOLO algorithm has been applied in some aspects but there are still problems in applying it to the action recognition of teachers and students in the classroom. The main problems are: (1) When the light is dim and the lighting distribution is uneven, the collected data are not clear enough, and the detection accuracy cannot meet expectations. (2) Occlusion and semi-hidden situations between different students often occur, which

can easily lead to missed detection and the need for a higher generalization ability of the model.

This paper proposes a classroom behavior recognition method based on improved YOLO-v4 to identify the classroom behaviors of teachers and students, in order to provide a reference for classroom behavior research. The main contributions are listed as follows:

(1) In view of the outstanding problems of teacher–student behavior detection in the sustainable classroom teaching process, embedded connection (EC) components are designed and successfully embedded in the end of the neck network of YOLO-v4 to enhance the learning and generalization ability of the network;

(2) By introducing the Repulsion loss function, the original loss function of YOLO-v4 is improved and optimized to reduce the number of false positive and false negative samples, thereby alleviating the problem that it is difficult to effectively identify the behaviors of targets due to mutual occlusion among students.

## 2. Related Research

Object detection is a key research direction in computer vision. Through the steps of image collection, image preprocessing, relevant feature extraction and result classification, the identification and classification of different targets such as pedestrians, vehicles and furniture items can be performed under various research fields [13].

T. Ojala et al. [14] proposed the LBP (Local Binary Pattern) operator to describe the characteristics of the local textures of the images. Paul Viola et al. [15] proposed the Viola-Jones algorithm and four Haar-like features, and then Adaboost classifier and Cascade classifier were combined with Haar-like features to achieve efficient and accurate image target detection and localization. Dalal et al. [16] proposed the HOG (Histogram of Oriented Gradient) detection algorithm, which had an excellent performance in pedestrian detection. Felzenszwalb et al. [17] proposed the DPM (Deformable Part Model) algorithm, which improved the HOG feature and combined the sliding window technique and the SVM classifier.

In recent years, the mainstream research methods of computer vision have shifted from the traditional manual feature-based classifier learning and classification models to "end-to-end" learning that combines machine learning and deep learning theories and generates and learns features autonomously. Geoffrey et al. [18] proposed the deep learning model AlexNet and achieved a huge advantage in the ImageNet Large Scale Visual Recognition Challenge. Ross Girshick et al. [19] proposed the R-CNN detection algorithm, which adopts the two-stage concept. First, the candidate regions are generated, and then the candidate regions are regressed and trained to obtain the bounding boxes with corresponding confidence. Later, more advanced versions such as Fast-RCNN [20] and Faster-RCNN [21] were successively proposed, which improved the training speed and accuracy.

Joseph Redmon et al. [11] proposed the YOLO-v1 model, in which the one-stage concept is adopted, the object categories and the bounding boxes are predicted at the same time, which makes the training speed of the yolo algorithm quite fast, but the accuracy rate is slightly reduced. After that, Joseph Redmon proposed improved algorithms such as YOLO-v2 [22], YOLO-v3 [23] and YOLO-v4 [12], which solved the problem of difficulty in small target recognition. In addition, multiple targets can be detected in a single grid, which achieves better target detection effect. Wei Liu et al. [24] proposed the SSD (Single Shot MultiBox Detector) detection algorithm, which, like the yolo series, can predict object categories and bounding boxes at the same time.

Behavior classification is an important direction in computer vision. For a specified video clip of human activities, the behaviors of the characters are recognized and classified. Shreyank et al. [25] proposed a human behavior classification method combined with a deep belief network, which is divided into two networks, one of which adopts an improved weber descriptor to obtain features from target motions. Another network is used to extract features from images, where the temporal and spatial information of the actions

in the frames is encoded to extract the corresponding features, which are then passed to the CNN for classification, resulting in an excellent classification performance. Tran et al. [26] first proposed to use the three-dimensional neural network (3D-NN) instead of the two-dimensional neural network in video action recognition tasks. On this basis, Hara et al. [27] proposed a three-dimensional network-based residual neural network (ResNet). Qiu et al. [28] proposed a "pseudo" 3D convolutional residual network (P-3D) that simulates SD-NN in two dimensions. Carreira et al. [29] proposed a two-stream inflated 3D ConvNet (I-3D), which uses the 2D network weight expansion as the pre-training weight of the 3D network. At the same time, other effective video action recognition methods are constantly being explored, such as the recognition framework based on long short-term memory (LSTM) and the video action recognition framework based on generative adversarial neural network (GAN) [30]. Guo et al. [31] proposed a new method to identify students' head raising rate (HRR) in the classroom and developed a method to extract the salient facial features of students. By constructing a multi-task CNN, the HRR of students can be detected, and the effectiveness of the method was proved through experiments. Jisi A et al. [32] proposed to combine spatial affine transformation network with CNN to extract more detailed features. Then, the weighted summation method is used to fuse the spatiotemporal features, and the classification and recognition are carried out through the improved softmax classifier. Abdallah et al. [33] proposed a new method based on deep transfer learning, which first pre-trained the model on the facial expression dataset, and then used the transfer model to classify students' behaviors. Liu et al. [34] proposed an improved YOLO-v3 network, which introduced a cascaded improved RFB module to enhance the feature extraction capability of the original network and make full use of shallow information to improve the recognition effect of small targets. Aiming at the problem of character occlusion caused by the classroom structure and the density of students, the resn module of Darknet-53 in YOLOv3 is replaced with the SE-Res2net module to realize the reuse of multi-layer features.

## 3. Classroom Behavior Recognition Model

### 3.1. YOLO-v4 Network

The YOLO (You Only Look Once) series are some of the most widely used one-stage target detection algorithms, and YOLO-v4 is the most popular version at present. The YOLO-v4 model mainly consists of four basic components: Input layer, Backbone network, Neck network and Output layer. The input layer takes fixed-size images, extracts features through the backbone network and then sends them to the neck network for feature aggregation. Finally, the output layer (YOLO Head) outputs the predicted anchor boxes of three different scales.

#### 3.1.1. Backbone Network

The YOLO-v4 architecture uses CSPDarknet53 as the backbone network, which is proposed on the basis of Darknet53 with reference to the cross-stage partial network CSP-Net [35]. Darknet53 is the backbone of YOLO-v3, so named because it contains 53 convolutional layers. YOLO-v4 retains the framework of Darknet53 but adopts the CSP (cross-stage partial network) modules to optimize the gradient back-propagation path, and at the same time greatly reduces the amount of calculation while ensuring the detection accuracy. The basic structure of CSPDarknet53 is given in Figure 1, where CBM stands for convolutional block and CSP refers to the cross-stage partial network.

As shown in Figure 1, CSPDarknet53 has two main functional modules: (1) CBM (Conv + BN + Mish), which is used to control stitching and sampling, where Mish is a new activation function used in YOLO-v4 (instead of Leaky_relu in YOLO-v3); (2) CSP, a set of cross-stage residual units (eg. CSP8 consists of eight residual units). The main difference between the two models is that YOLO-v4 refers to the CSPNet mechanism in which the connection structure of the residual units is improved and a bypass across the residual blocks is added to form a cross-stage local connection; thus, the risk of gradient

disappearance is effectively reduced. Simply stated, in the backbone network of YOLO-v4, a residual unit can form two paths: one is directly connected to its next residual unit; the other is connected to the residual unit at the end of the CSP (bypass). Due to the addition of a bypass in YOLO-v4, the reasoning of CSP is more reasonable, and the internal gradient flow is effectively divided, reducing the risk of gradient disappearance, thereby enhancing the generalization of network learning.

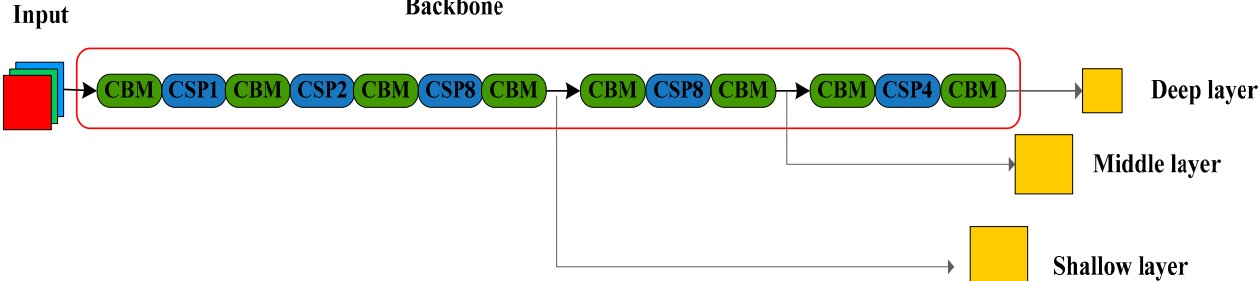

**Figure 1.** The backbone network structure of YOLO-v4.

### 3.1.2. Neck Network

The Neck network is a feature transfer network added between the backbone network and the prediction output layer. Its main function is to sample and aggregate the feature values extracted from the backbone network to form aggregated features at different scales. The Neck network of YOLO-v4 adopts the PANet (Path Aggregation Network) structure [36]. As shown in Figure 2, the feature pyramid and path aggregation techniques are used in PANet, which makes the information of the lower layer easier to spread to the upper part, and the positioning is more accurate, while supporting the prediction of large, medium and small targets.

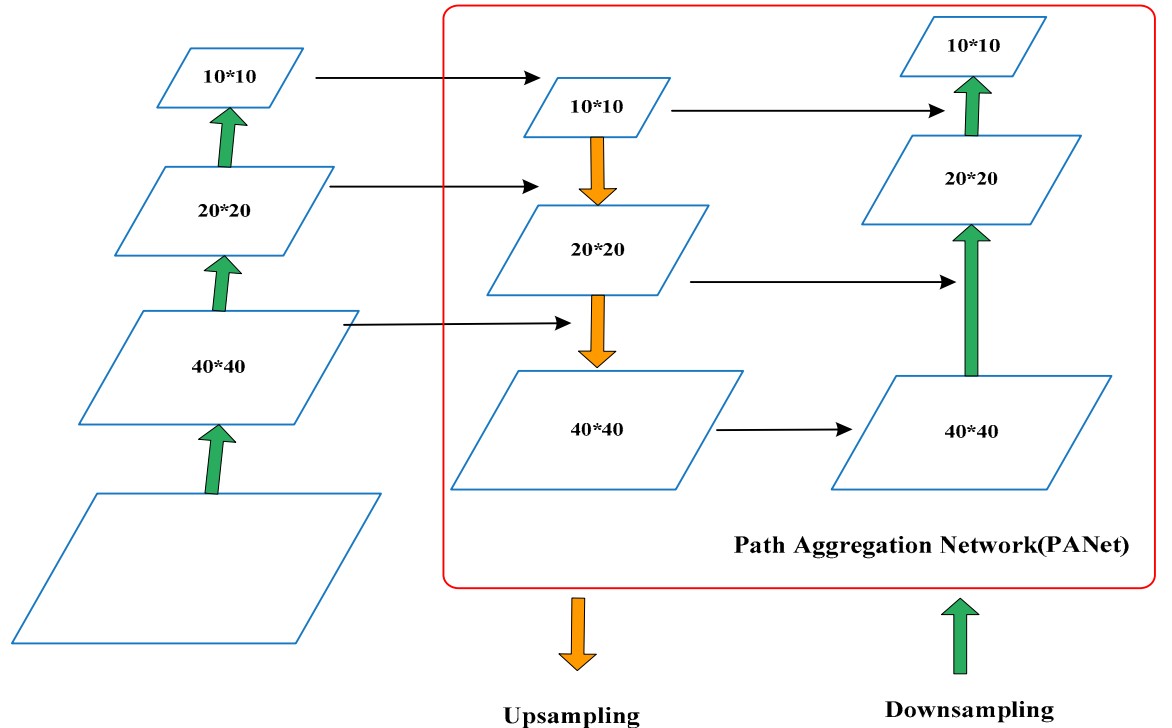

**Figure 2.** PANet structure.

### 3.2. EC Component Design

The purpose of designing EC components is mainly to enhance the adaptability and generalization ability of feature aggregation, thereby effectively improving the performance of feature aggregation. The block diagram of the EC component is shown in Figure 3. As shown in Figure 3a, there are two paths in the EC module, one consists of two units, namely conv1 and conv2, and the other contains only conv2. In this way, combining the results of the two pathways can effectively alleviate the bias in the aggregation process, and using different excitation functions for conv1 and conv2 also increases the adaptability of the EC component. As shown in Figure 3b, the convolutional unit conv1 uses the Leaky_Relu excitation function which retains the characteristics of the original YOLO-v4. It can be found from Figure 3c that the convolutional unit con2 uses the Linear excitation function. From a prediction perspective, Linear functions are faster and easier to produce predictable outputs. In this paper, the EC components are incorporated in the YOLO-v4 network, that is, the EC components are added at the end of the Neck network to form an improved YOLO-v4 network.

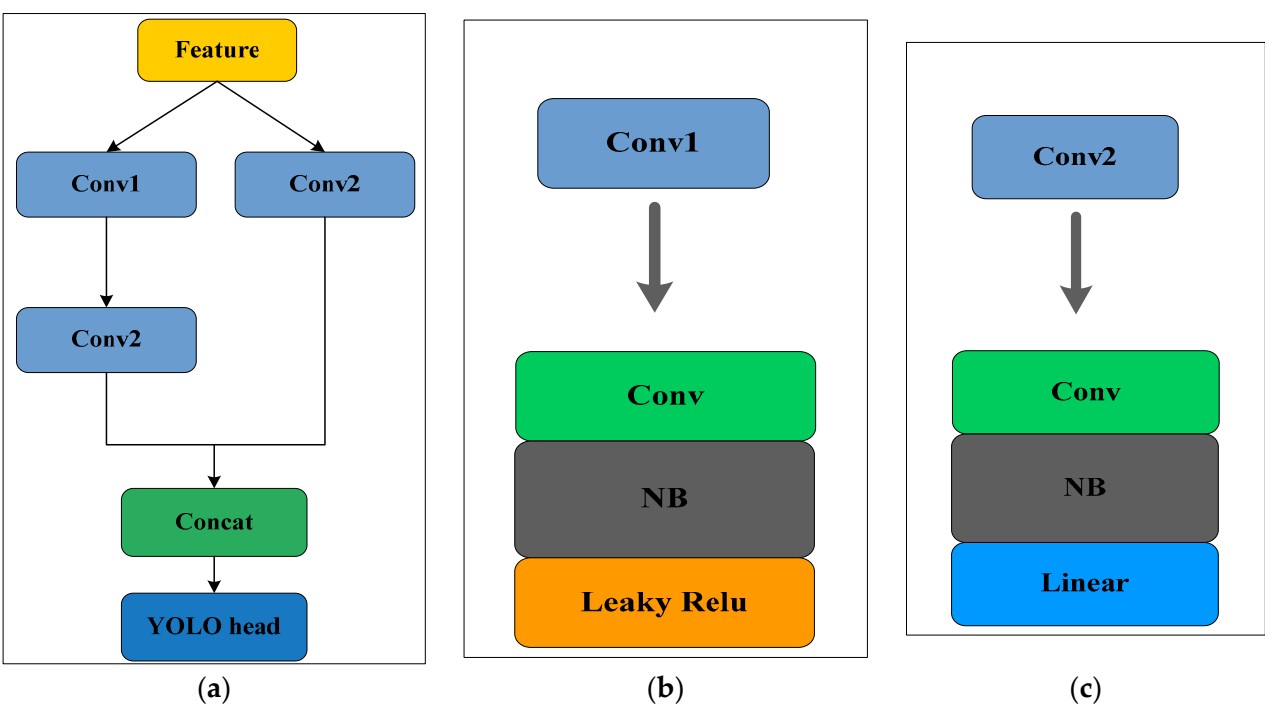

**Figure 3.** EC component. (**a**) structure of EC; (**b**) Convolution 1; (**c**) Convolution 2.

### 3.3. Model Design

The improved YOLO-v4 network is obtained by adding EC components at the end of the Neck network. In the YOLO-v4 network with EC components embedded, the Neck network has better back-propagation ability. The end of the Neck network is directly connected to the prediction layer of YOLO-v4, and the enhanced back-propagation ability helps to improve the generalization ability of the network and reduce prediction bias. At the same time, since the EC components are embedded at the end of the entire network, shallow aggregated information is added to the final prediction, which can effectively improve the accuracy of the prediction. Figure 4 shows the detailed structure diagram.

It can be found from the structure of Figure 4 that the feature extraction stage is based on the CSPDarknet53 of the original YOLO-v4. The main change is that in the feature aggregation stage, the network extracts features at three scales: Firstly, the deep information is up-sampled twice and the shallow information is aggregated. The aggregated information is then down-sampled twice to make predictions at the scales of $40 \times 40$, $20 \times 20$, and $10 \times 10$ pixels, which correspond to large, medium and small-sized targets,

respectively. The deeper information is responsible for detecting large-sized objects, and the shallower information is responsible for detecting small-sized objects, thereby improving the adaptability of the network.

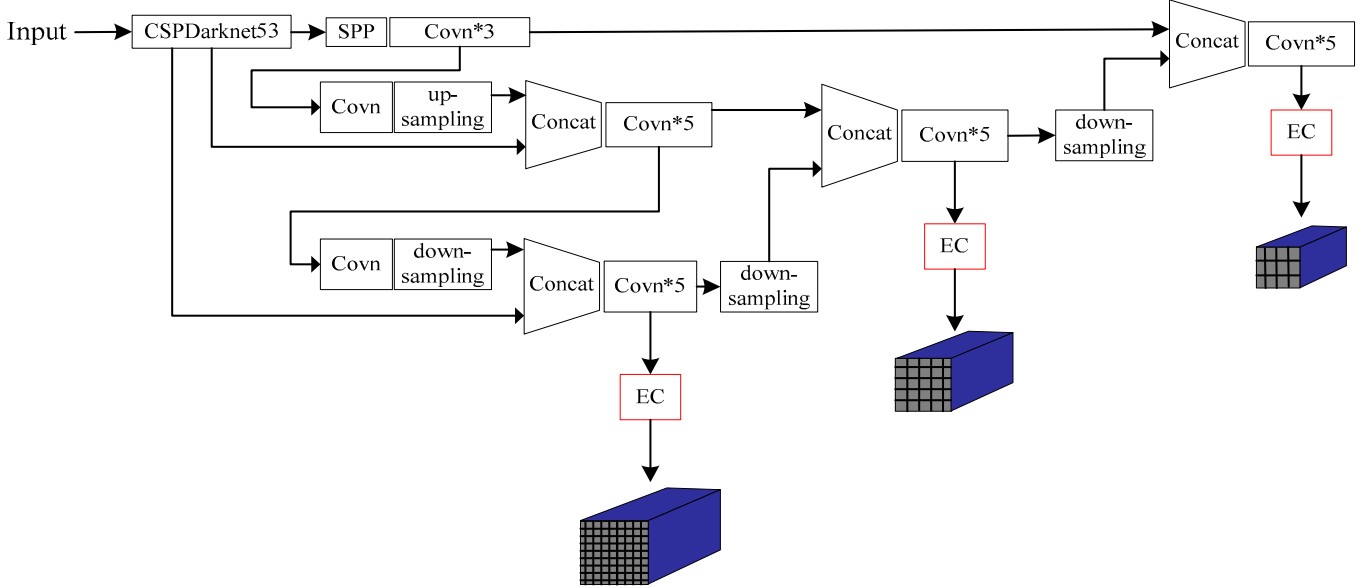

**Figure 4.** Improved network structure.

*3.4. Loss Function Modification*

The loss function of the original YOLO-v4 algorithm consists of three parts: bounding box regression loss $L_{\text{CIoU}}$, confidence loss $L_{\text{con}}$ and classification loss $L_{\text{class}}$, which can be expressed as:

$$L = L_{\text{CIoU}} + L_{\text{con}} + L_{\text{class}} \tag{1}$$

where the bounding box regression loss function can be defined as:

$$L = L_{\text{CIoU}} = 1 - S_{\text{IoU}} + \frac{\rho^2(b, b_1)}{c^2} + v\alpha \tag{2}$$

$$v = \frac{4}{\pi^2} \left( \arctan \frac{w_1}{h_1} - \arctan \frac{w}{h} \right)^2 \tag{3}$$

$$\alpha = \frac{v}{(1 - S_{\text{IoU}}) + v} \tag{4}$$

where $S_{\text{IoU}}$ is the intersection over union (IoU) of the areas between the ground truth bounding box and the predicted bounding box. $b$ and $b_1$ are the centroid coordinates of the predicted bounding box and the ground truth bounding box, respectively; $\rho^2(b, b_1)$ is the Euclidean distance between the centroid coordinates of the predicted bounding box and the ground truth bounding box. $c^2$ is the diagonal distance of the smallest closed area between the predicted bounding box and the real bounding box. $w_1$ and $h_1$ are the width and height of the ground truth bounding box, respectively; $w$ and $h$ are the width and height of the predicted bounding box, respectively. $v$ is the similarity measure of the aspect ratio. If the ground-truth bounding box and the predicted bounding box have similar widths and heights, then $v$ is 0. $\alpha$ is the weight function.

The confidence loss function $L_{con}$ and the class loss function $L_{class}$ are calculated as follows:

$$L_{con} = -\sum_{i=0}^{K \times K} \sum_{j=0}^{M} I_{ij}^{obj} [\hat{c}_{ij} \log(c_{ij}) + (1 - \hat{c}_{ij}) \log(1 - c_{ij})] - \lambda_{noobj} \sum_{i=0}^{K \times K} \sum_{j=0}^{M} I_{ij}^{obj} [\hat{c}_{ij} \log(c_{ij}) + (1 - \hat{c}_{ij}) \log(1 - c_{ij})] \tag{5}$$

$$L_{class} = -\sum_{i=0}^{K \times K} \sum_{j=0}^{M} I_{ij}^{obj} \sum_{c \in classes} [\hat{P}_{ij}(c) \log(P_{ij}(c)) + (1 - \hat{P}_{ij}(c)) \log(1 - \hat{P}_{ij}(c))] \qquad (6)$$

where $K$ represents the size of the feature map; $M$ is the number of priori bounding boxes used by the feature map; $c_{ij}$ is the confidence level, and $c$ is the predicted category.

### 3.5. Improved Loss Function

The limbs of students in the classroom are often occluded. If the YOLO-v4 model is directly used for identification, it is prone to false detection and missed detection. In order to solve the problem of false detection and missed detection in the recognition process, we combined the Repulsion loss function with the original YOLO-v4 loss function to achieve a better performance. The loss function from the original YOLO-v4 and the Repulsion loss function were concatenated to obtain the improved loss function:

$$L_R = L + \alpha L_{RepGT} + \beta L_{RepBox} \qquad (7)$$

where $L$ is the original loss function of YOLO-v4. $L_{RepGT}$ is the loss value generated between the predicted bounding box and the adjacent ground-truth bounding boxes of the target; $L_{RepBox}$ is the loss value generated between the predicted bounding box of the target and other adjacent predicted bounding boxes of other targets. The weight coefficients $\alpha$ and $\beta$ are used to balance the two loss values.

The RepGT loss function is calculated as:

$$L_{RepGT} = \frac{\sum_{P \in P_1} \text{Smooth}_{ln}(I_{IoG}(B_P, G_{Rep}))}{|P_1|} \qquad (8)$$

$$I_{IoG}(B, G) = \frac{A_{area}(B \cap G)}{A_{area}(G)} \qquad (9)$$

$$\text{Smooth}_{ln} = \begin{cases} -\ln(1-x) & x \leq \delta \\ \frac{x-\delta}{1-\delta} - \ln(1-\delta) & x > \delta \end{cases} \qquad (10)$$

where $B_P$ is the predicted bounding box of the target obtained by the regression adjustment of the predicted bounding box $P$; $G_{Rep}$ is the ground truth bounding box with the largest IoU ratio with the predicted bounding box of the objects except the specified target. $P_1$ is the set of positive examples whose overlap level between the predicted bounding box $P$ and the ground truth bounding box reaches the threshold. $A_{Area}()$ is an area calculation function. The $\delta$ value in the Smoothln function is used to adjust the sensitivity of the entire loss function to the bounding box pair with larger intersection values. When $\delta = 1$, $L_{RepGT}$ achieves the best effect; when $\delta = 0$, $L_{RepBox}$ achieves the best effect. In this experiment, we took $\delta = 0.5$ based on the experience.

The RepBox loss function $L_{RepBox}$ can be calculated as:

$$L_{RepBox} = \frac{\sum_{i \neq j} Smooth_{ln}(S_{IoU}(B_{P_i}, B_{P_j}))}{\sum_{i \neq j} 1[S_{IoU}(B_{P_i}, B_{P_j}) > 0] + \varepsilon} \qquad (11)$$

where $P_i$ and $P_j$ are the predicted bounding boxes of different targets. $B_{P_i}$ and $B_{P_j}$ are the predicted bounding boxes of the targets regressed from the predicted bounding boxes $P_i$ and $P_j$, respectively. $S_{IoU}(B_{P_i}, B_{P_j})$ is the IoU ratio between $B_{P_i}$ and $B_{P_j}$. $\varepsilon$ is a minimum value set to prevent the divisor from being 0. The indicative function in the denominator indicates that the loss value is only calculated for the prediction bounding box pairs with an intersection, otherwise the loss calculation is not performed.

It can be seen from Equation (8) that when the intersection between $B_P$ and $G_{Rep}$ is large, that is, the closer the two are, the larger the RepGT loss is, which makes the predicted bounding box of the target and the adjacent ground-truth bounding boxes of the targets far away, so the false detection of the target can be effectively prevented. It can be seen from

Equation (11) that when the intersection of $B_{P_i}$ and $B_{P_j}$ is large, that is, the closer the two are, the larger the RepBox loss is, and the predicted bounding box of the target will be far away from the adjacent predicted bounding boxes of other similar objects, reducing the probability that the predicted bounding boxes of different targets are mistakenly deleted after non-maximum suppression, thereby alleviating the problem of missed detection of targets.

### 3.6. Selection Principle of Teacher–Student Behavior Set

The raw data used in this experiment were about 500 min of classroom teaching videos shot on the spot. The selected behaviors had obvious characteristics, and there can be no ambiguity or overlap, so as to facilitate the task of behavior recognition. The statistical information of the dataset is shown in Table 1. The selected images were manually labeled using LabelImg software, of which the training set accounted for 80% and the test set accounted for 20% of the whole dataset.

**Table 1.** Statistics of the Experiment Dataset.

| Student Behavior | Look UP | Head Drop | Hand Raise | Stand Up | Lying on the Desk | | | |
|---|---|---|---|---|---|---|---|---|
| Frames # | 3512 | 1432 | 7011 | 680 | 912 | | | |
| Teacher behavior | Explain questions | Pointing to the projection | No hand gestures | Gesture with both hands | Head down and operate | Walk around | Blackboard-writing | Guide to raise hand |
| Frames # | 211 | 1354 | 6985 | 5012 | 515 | 8098 | 12,175 | 7121 |

The training parameters of the network were as follows: in the experiment, the training iteration was set to 100 rounds, the learning rate and batch size of the first 50 rounds of training were 0.001 and 16, respectively, and the training rate and batch size of the last 50 rounds of training were 0.0001 and 8, respectively. The momentum factor was set to 0.9. The batch size represents the number of samples per input to the network for training; the momentum factor represents the decreasing trend of the value of the loss function during training.

## 4. Experiment

### 4.1. Experimental Environment

The experimental hardware platform was configured with Intel® CoreTM i7-8700 CPU, 8 G memory, GeForce GTX 1650 4G, with Windows10 operating system. The raw data of the teaching video came from the smart classroom deployed by China Hangzhou Hikvision Digital Technology Co., Ltd. The smart classroom was a multi-functional classroom with multimedia equipment. There were 2 cameras above the classroom: the camera in the middle (iDSEGD0288- HFR (2.8 mm)) recorded the behaviors of the teacher on the podium; the camera in the front of the classroom (iDS-ECD8012- H/T (8.0~32.0 mm)) was used to record the behaviors of students in the classroom. The cameras deployed in these two locations were used to record videos, which were used as the sources of teacher behavior and student behavior datasets, respectively, as shown in Figure 5.

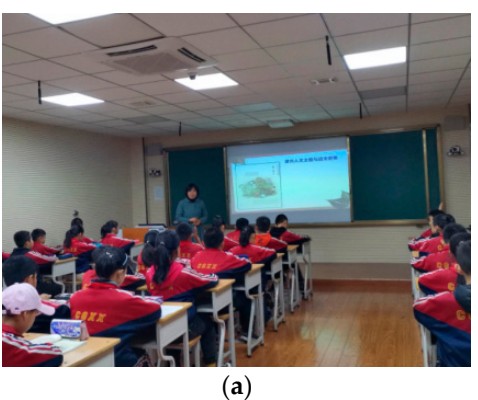 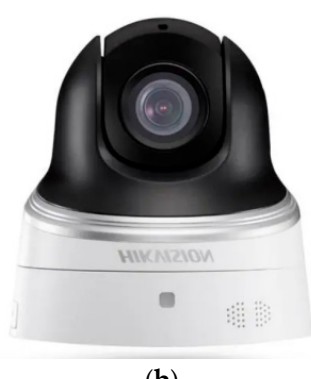 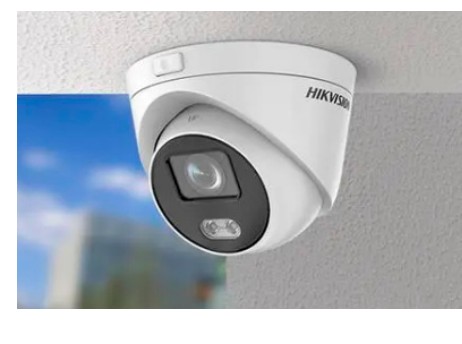

(**a**)          (**b**)          (**c**)

**Figure 5.** Experimental environment and equipment. (**a**) Smart class; (**b**) iDSEGD0288-HFR; (**c**) iDS-ECD8012-H/T.

### 4.2. Evaluation Metrics

Precision (*P*), recall rate (*R*) and average precision (*AP*) were used as performance metrics in the experiments. Precision calculated the proportion of correctly predicted positive samples in the total number of predicted positive samples. The recall rate calculated the proportion of correctly predicted positive samples in the total number of actual positive samples:

$$P = \frac{TP}{TP + FP} \tag{12}$$

$$R = \frac{TP}{TP + FN} \tag{13}$$

where *TP* is the number of correctly classified positive samples, *FP* is the number of predicted positive samples that were actually negative samples and *FN* is the number of predicted negative samples that were actually positive samples.

The *AP* was used as an evaluation index to measure the performance of the improved algorithm. Generally, the better the performance of the target recognition algorithm, the higher the value of *AP*. AP represents the average recognition accuracy of the network for each type of target. It is calculated as the area of the image enclosed by the precision–recall curve of each type of target and the *x*-axis:

$$AP = \int_0^1 P(R)d(R) \tag{14}$$

In the evaluation of processing speed, the number of frames per second (frames/s) was used as the evaluation index for determining the detection speed.

### 4.3. Results Analysis

The proposed behavior recognition algorithm under teaching scenario was evaluated on the experimental dataset, and the confusion matrix results are shown in Figure 6. A confusion matrix is a specific matrix used to visualize the performance of a deep learning algorithm. Each column represents the predicted results and each row represents the ground-truth fact. In the confusion matrix shown in Figure 6, the values on the diagonal represent the accuracy of the model predictions for different categories.

It can be seen from Figure 6 that the proposed model achieved relatively accurate results for the behavior recognition of both teachers and students. Among the recognition results of different teacher behaviors shown in Figure 6a, the recognition accuracy of walking was the lowest, because walking is an action that contains time information, so it is more difficult to recognize than other behaviors. Among the recognition results of different behaviors of students shown in Figure 6b, the behavior of hand raising had the lowest recognition accuracy. This is because additional actions cheek-holding and head-scratching

that often occur in class may be mistakenly recognized as hand raising actions, leading to a relatively low recognition accuracy.

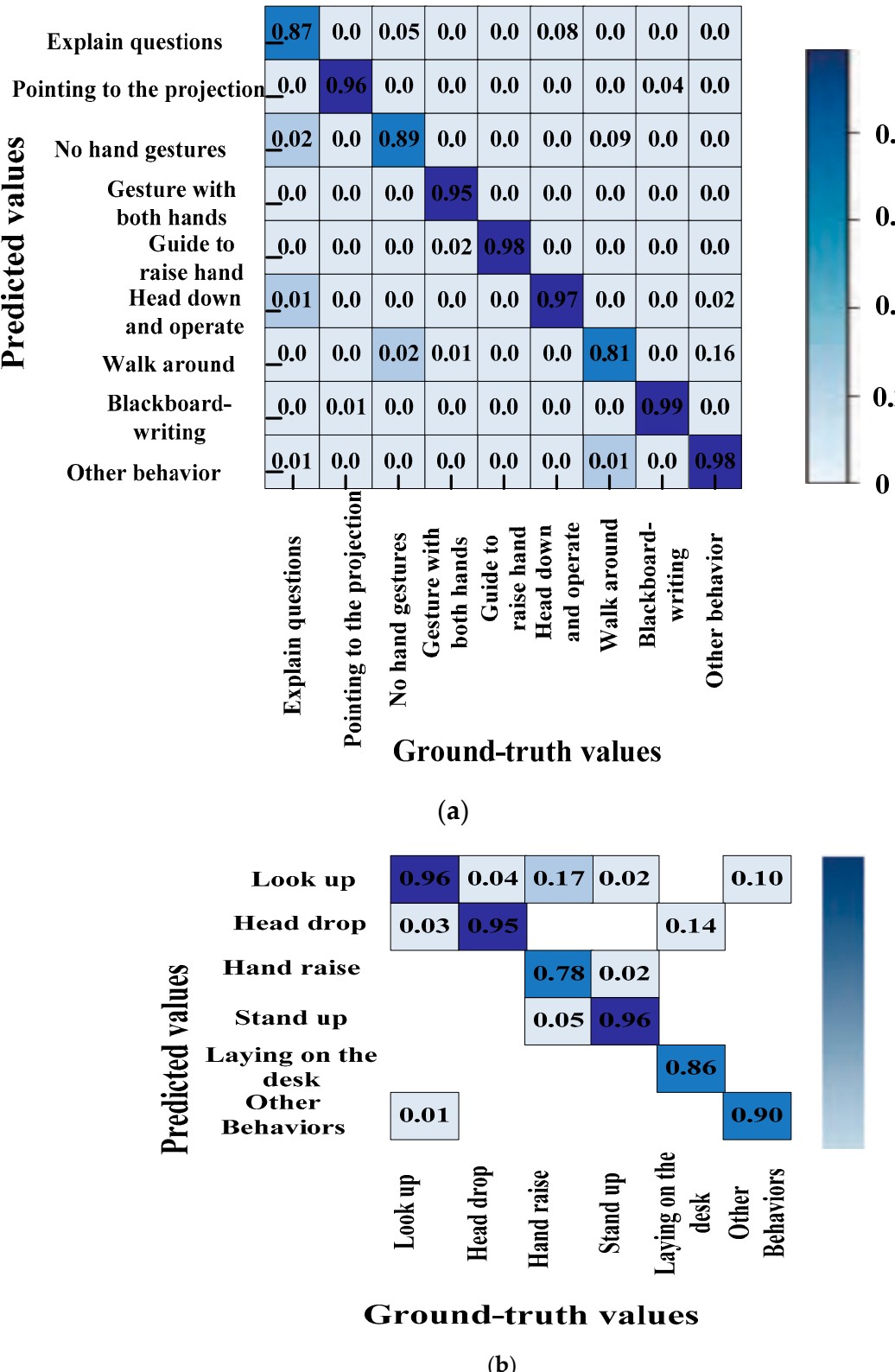

**Figure 6.** Results in the confusion matrix. (**a**) Teacher behavior recognition results; (**b**) student behavior recognition results.

The evaluation results of classroom behavior recognition for teachers and students are shown in Tables 2 and 3, respectively. It can be seen from Tables 2 and 3 that the proposed behavior recognition framework achieved good results in identifying teachers and students' classroom behaviors, both in terms of precision and recall.

**Table 2.** Evaluation results on the teacher dataset.

| Behavior | Explain Questions | Pointing to the Projection | No Hand Gestures | Gesture with Both Hands | Head Down and Operate | Walk Around | Blackboard-Writing | Guide to Raise Hand |
| --- | --- | --- | --- | --- | --- | --- | --- | --- |
| Precision | 0.912 | 0.955 | 0.941 | 0.985 | 0.936 | 0.951 | 0.990 | 0.979 |
| Recall | 0.909 | 0.963 | 0.922 | 0.969 | 0.911 | 0.903 | 0.939 | 0.960 |

**Table 3.** Evaluation results on the student dataset.

| Behavior | Look Up | Head Drop | Hand Raise | Stand Up | Lying on the Desk |
| --- | --- | --- | --- | --- | --- |
| Precision | 0.970 | 0.959 | 0.941 | 0.982 | 0.899 |
| Recall | 0.957 | 0.938 | 0.894 | 0.979 | 0.905 |

### 4.4. Performance Comparison

In order to verify the convergence speed and stability of the improved YOLO-v4 algorithm, the loss function curves of the improved algorithm and the other state-of-the-art algorithms including YOLO-v4 [12], YOLO-v3 [23] and Faster RCNN [21] were compared. In order to compare the convergence effect more clearly, the total loss value of each round was selected as the comparison value, and the recognition results on the student behavior dataset are shown in Figure 7. It can be seen that all the comparison algorithms finally reached convergence after training, but the improved YOLO-v4 algorithm had a faster convergence speed and a smoother loss function curve compared with the other three algorithms. Moreover, after 85 rounds of training, the proposed algorithm obtained the smallest loss value, and the fluctuation range of the loss value was also smaller. The results show that the improved YOLO-v4 algorithm proposed in this paper had a better robustness while ensuring the convergence performance.

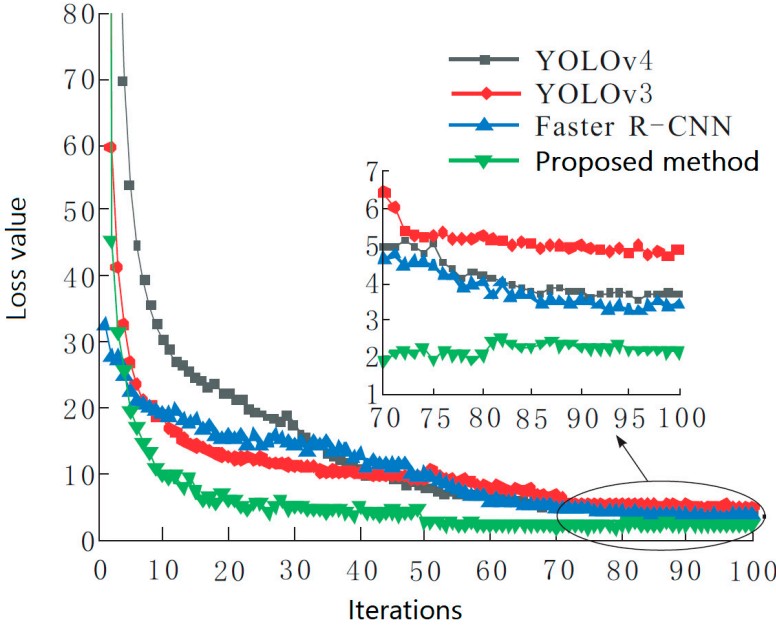

**Figure 7.** Convergence performance comparison.

In order to verify the performance of the proposed algorithm, the improved YOLO-v4 algorithm was compared with the YOLO-v4, YOLO-v3 and Faster R-CNN algorithms in terms of AP and recognition speed on the student behavior dataset, and the results are listed in Table 4. It can be seen from the results in the table that the AP of student behavior recognition when using the proposed improved YOLO-v4 algorithm was 97.85%, which was 2.43%, 3.78% and 5.95% higher than that of YOLO-v4, YOLO-v3 and Faster R-CNN, respectively. In terms of recognition speed, the proposed algorithm was faster than the other three algorithms, reaching 58 frames/s. In general, the proposed improved YOLO-v4 algorithm improved the accuracy of behavior recognition and sped up the speed of action detection, thus realizing the fast and effective detection of classroom behaviors.

**Table 4.** Comparison results on the student behavior dataset.

| Models | Backbone Network | AP/% | Speed (Frame/s) |
|---|---|---|---|
| YOLO-v4 [12] | CSP-Darknet53 | 95.42 | 41 |
| YOLO-v3 [23] | Darknet53 | 94.07 | 33 |
| Faster R-CNN [21] | VGG16 | 91.89 | 12 |
| Ref. [31] | YOLO-v3 | 90.57 | 35 |
| Ref. [32] | Feature fusion network | 92.16 | 10 |
| Proposed method | CSP-Darknet53 | 97.85 | 58 |

Finally, Figure 8 shows the detection examples of the teaching video in the real scene after key frame extraction. It can be found from the figure that in the real teaching scenario, the proposed model had good generalization ability in the task of student behavior recognition and could complete the recognition task accurately.

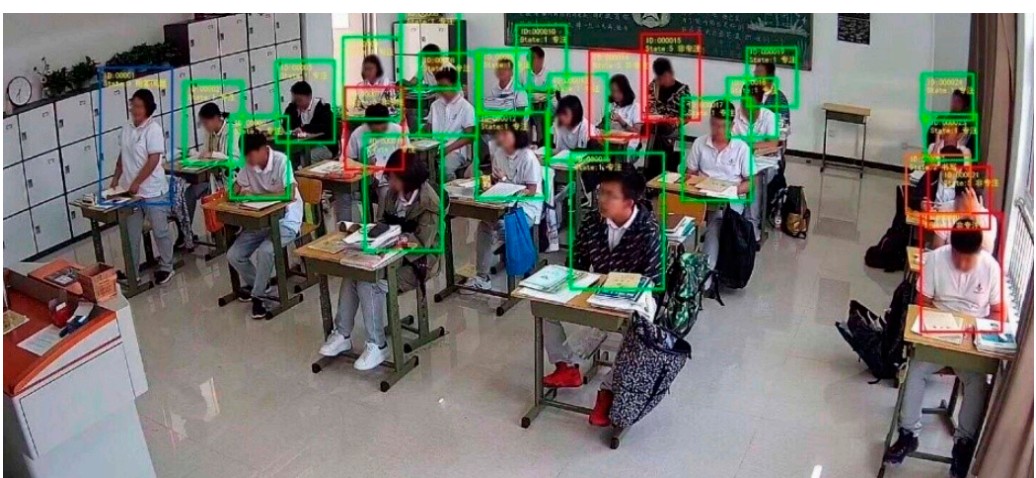

**Figure 8.** Visualization result illustration.

## 5. Conclusions

With the development of CNN (Convolutional Neural Networks) and IoT (Internet of Things) technologies, target detection algorithms based on deep learning have become mainstream, and typical algorithms such as SSD (Single Shot Detection) and YOLO series have emerged. Based on the videos or images collected in the perception layer of the IoT paradigm, deep learning models are used in the processing layer to implement various intelligent applications. In this paper, a framework for identifying sustainable classroom behaviors of teachers and students based on an optimized deep learning model was proposed, and the effectiveness of the proposed method was verified through comparative experiments. The two datasets of sustainable classroom behaviors of teachers and students were trained and tested, respectively. On the self-constructed student sustainable classroom behavior dataset, the average recognition accuracy of identifying various sustainable classroom behaviors was over 90%, which verified the effectiveness of the method. In the

future, we will improve the generalization of sustainable classroom behavior recognition methods and try to expand the datasets of teacher and student behaviors to provide more diverse data for smart sustainable classroom analysis.

**Author Contributions:** Conceptualization, methodology, writing—original draft preparation, writing—review and editing, H.C. and J.G. All authors have read and agreed to the published version of the manuscript.

**Funding:** This work was supported by Key Project of Natural Science Foundation of Fujian Province, China, "Research on Key Technology of Intelligent Inspection Robot in Power Distribution Room". (No.2020J02048), Project Leader: Jiansheng Guan.

**Data Availability Statement:** Data are contained within the article.

**Conflicts of Interest:** The authors declare no conflict of interest.

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
