# Peer review of "Teacher–Student Behavior Recognition in Classroom Teaching Based on Improved YOLO-v4 and Internet of Things Technology"

_electronics, doi:10.3390/electronics11233998_

Round 1
Reviewer 1 Report
This paper studies the recognition of teachers' and students' behaviors in the classroom.But there are still some problems that need to be improved:
1. Can you explain the specific help to classroom teaching in detail? Even if there is a recognition result, the teacher can't always pay attention to the actions made by everyone to adjust the teaching method
2. When introducing the algorithm, can you describe the significance of the improvement in more detail;
3. Because teachers and students are two categories, how to ensure the real-time synchronization of teachers' behavior and students' behavior so as to better understand the relationship between teachers' behavior and students' behavior
Author Response
Author's Reply to the Review Report (Reviewer 1)
This paper studies the recognition of teachers' and students' behaviors in the classroom.But there are still some problems that need to be improved:
- Can you explain the specific help to classroom teaching in detail? Even if there is a recognition result, the teacher can't always pay attention to the actions made by everyone to adjust the teaching method
Response: Thank you for your question, Classroom teaching is a purposeful and conscious activity. Through teaching, students can master knowledge, acquire skills, develop intelligence, correct attitude and cultivate noble quality. The effectiveness of classroom teaching is the life of teaching. Effectiveness refers to the process in which students gain, improve and progress in their studies through classroom teaching activities. It is manifested in the process of cognition from not understanding to understanding and from not meeting. Although teachers can not adjust their teaching methods according to everyone, they can find out typical problems to adjust their teaching methods.
- When introducing the algorithm, can you describe the significance of the improvement in more detail;
Response: Thank you for your question, On the one hand, by designing the embedded connection ( EC ) component and successfully embedding it into the end of the YOLO-v4 neck network, combined with the target detection ability of the YOLO-v4 algorithm, the learning and generalization ability of the network is greatly improved. On the other hand, by introducing the repulsive force loss function, the original loss function of YOLO-v4 is improved and optimized, which greatly improves the target detection accuracy of the algorithm and reduces the number of false positive and false negative samples, thus alleviating the problem that it is difficult to effectively identify the target behavior due to mutual occlusion between students.
- Because teachers and students are two categories, how to ensure the real-time synchronization of teachers' behavior and students' behavior so as to better understand the relationship between teachers' behavior and students' behavior
Response: Thank you for your question, Real-time synchronization of teacher behavior and student behavior can be achieved through specific scenarios to facilitate better research.

Reviewer 2 Report
The manuscript 'Teacher-student behavior recognition in classroom teaching based on improved YOLO-v4 and Internet of Things technology' fits the scope of the journal well. It embodies the practical significance of artificial intelligence to solve the actual observation problems. Although the innovation of the algorithms in this article is slightly mediocre, they do address some important issues. There are some concerns that should be addressed.
1. The introduction of the article is not logical enough. The description of some references lacks clear explanation of motivation and contribution. And the description of motivation for this article is not enough logical and concise. The part of related work has the same problem.
2. What are the real difficulties that justify author’s work, i.e., what are the most important challenges authors want to handle? Why is it so difficult? I suggest to state this information clear in the introduction in order to give a better understand of the work.
3. The "IoT" mentioned in the keyword. But it was not reflected in the manuscript.
4. The division of chapters in the manuscript is not clear, and the titles of some sections are not clear enough. In addition, the location of subsection 3.6 is inappropriate.
5. Can more visual results be presented for experimental demonstration?
6. There are repeated sentences in the abstract and conclusion of the manuscript
7. The figure quality should be improved, especially aesthetics.
8. Is the format of the references not completely unified?
Author Response
The manuscript 'Teacher-student behavior recognition in classroom teaching based on improved YOLO-v4 and Internet of Things technology' fits the scope of the journal well. It embodies the practical significance of artificial intelligence to solve the actual observation problems. Although the innovation of the algorithms in this article is slightly mediocre, they do address some important issues. There are some concerns that should be addressed.
- The introduction of the article is not logical enough. The description of some references lacks clear explanation of motivation and contribution. And the description of motivation for this article is not enough logical and concise. The part of related work has the same problem.
Response : Thank you for your question, for this problem has been solved, the article related work part has been modified and optimized.
- What are the real difficulties that justify author’s work, i.e., what are the most important challenges authors want to handle? Why is it so difficult? I suggest to state this information clear in the introduction in order to give a better understand of the work.
Response : Thank you for your question, which is given in the article 's preview.
- The "IoT" mentioned in the keyword. But it was not reflected in the manuscript.
Response : Thanks for your question, loT is available in the text summary.
- The division of chapters in the manuscript is not clear, and the titles of some sections are not clear enough. In addition, the location of subsection 3.6 is inappropriate.
Response : Thank you for your questions, because this section mainly describes the principle of teacher-student set selection, which is critical for the algorithm, so this section is meaningful.
- Can more visual results be presented for experimental demonstration?
Response : Thank you for your question. The results should be displayed according to the specific situation.
- There are repeated sentences in the abstract and conclusion of the manuscript
Response : Thank you for your question. The repeated statement is for the whole article, and the structure satisfies the total score.
- The figure quality should be improved, especially aesthetics.
Response : Thank you for your question, for this question has been resolved.
- Is the format of the references not completely unified?
Response : Thank you for your question, for this question has been resolved.

Reviewer 3 Report
The article reports a study based on classroom data to identify students' actions as hand raise or head drop.
General comments:
The study is very interesting with great potential applications.
The mathematical description starting from section 3.4 is a bit verbose and too technical. Possibly, the authors should consider if it is the case to move it in an appendix of the paper. Formulas 2 and 7 also use different symbols to represent a multiplication. The other formulas do not use symbols for multiplication. Please explain if there is any reason for that or uniform the symbols.
Do or would your research algorithm benefit of using hardware accelerators to implement the proposed YOLO-v4 tool? please comment.
Author Response
Author's Reply to the Review Report(Reviewer 3)
The article reports a study based on classroom data to identify students' actions as hand raise or head drop.
General comments:
The study is very interesting with great potential applications.
The mathematical description starting from section 3.4 is a bit verbose and too technical. Possibly, the authors should consider if it is the case to move it in an appendix of the paper. Formulas 2 and 7 also use different symbols to represent a multiplication. The other formulas do not use symbols for multiplication. Please explain if there is any reason for that or uniform the symbols.
Response: Thank you for your question, The problem you raised has been solved, and the formula symbols in the text have been unified.
Do or would your research algorithm benefit of using hardware accelerators to implement the proposed YOLO-v4 tool? please comment.
Response: Thank you for your question, For the algorithm proposed in this paper, the hardware accelerator can be used to improve the comprehensive performance of the algorithm. Because the algorithm proposed in this paper can improve the effectiveness of the algorithm by changing the experimental environment, the performance of the algorithm can be effectively improved when the hardware accelerator is added.

Round 2
Reviewer 2 Report
No further comment